# Microstructural and Mechanical Characteristics of Alkali-Activated Binders Composed of Milled Fly Ash and Granulated Blast Furnace Slag with µ-Limestone Addition

**DOI:** 10.3390/ma16103818

**Published:** 2023-05-18

**Authors:** Francisco Vázquez-Rodríguez, Nora Elizondo, Myriam Montes-González, Cristian Gómez-Rodríguez, Yadira González-Carranza, Ana M. Guzmán, Edén A. Rodríguez

**Affiliations:** 1Programa Doctoral en Ingeniería Física, Facultad de Ciencias Físico Matemáticas (FCFM), Universidad Autónoma de Nuevo León (UANL), San Nicolás de los Garza 66450, Nuevo León, Mexico; 2Facultad de Arquitectura (FARQ), Universidad Autónoma de Nuevo León (UANL), San Nicolás de los Garza 66450, Nuevo León, Mexico; 3Consejo Superior de Investigaciones Científicas, Instituto de Ciencias de la Construcción Eduardo Torroja, C/Serrano Galvache 4, 28002 Madrid, Spain; 4Facultad de Ingeniería, Universidad de Veracruz, Coatzacoalcos 96535, Veracruz, Mexico; 5Facultad de Ingeniería Mecánica y Eléctrica (FIME), Universidad Autónoma de Nuevo León, Av. Pedro de Alba S/N, San Nicolás de los Garza 66450, Nuevo León, Mexico

**Keywords:** slag, fly ash, µ-limestone, geopolymer, composite

## Abstract

Concrete is the most used construction material, needing large quantities of Portland cement. Unfortunately, Ordinary Portland Cement production is one of the main generators of CO_2_, which pollutes the atmosphere. Today, geopolymers are an emerging building material generated by the chemical activity of inorganic molecules without the Portland Cement addition. The most common alternative cementitious agents used in the cement industry are blast-furnace slag and fly ash. In the present work, the effect of 5 wt.% µ-limestone in mixtures of granulated blast-furnace slag and fly ash activated with sodium hydroxide (NaOH) at different concentrations was studied to evaluate the physical properties in the fresh and hardened states. The effect of µ-limestone was explored through XRD, SEM-EDS, atomic absorption, etc. The addition of µ-limestone increased the compressive strength reported values from 20 to 45 MPa at 28 days. It was found by atomic absorption that the CaCO_3_ of the μ-limestone dissolved in NaOH, precipitating Ca(OH)_2_ as the reaction product. SEM-EDS analysis showed a chemical interaction between C-A-S-H- and N-A-S-H-type gels with Ca(OH)_2_, forming (N, C)A-S-H- and C-(N)-A-S-H-type gels, improving mechanical performance and microstructural properties. The addition of μ-limestone appeared like a promising and cheap alternative for enhancing the properties of low-molarity alkaline cement since it helped exceed the 20 MPa strength recommended by current regulations for conventional cement.

## 1. Introduction

Portland cement (PC), a traditional cementitious material, is the most predominant binder used in construction with a large consumption worldwide [1]. However, cement manufacturing causes alarming greenhouse gas (GHG) emissions, raw materials consumption, and intensive energy expenses. The whole production process of PC consumes about 3.2 GJ energy per ton for raw material mining, transporting, clinker calcination, and grinding. Almost 810 kg of carbon dioxide, 1.0 kg of sulfur dioxide, and 2.0 kg of nitrogen oxides are generated per ton of cement manufactured [2,3,4]. Projections for the global demand for Portland cement predict there will be about six billion tons per year, generating over 4.8 billion tons of carbon dioxide (CO_2_). CO_2_, as the main greenhouse gas, is responsible for global warming, and the cement industry causes around 8% of CO_2_ emissions worldwide [5]. CO_2_ emissions from cement production are generated by the combustion essential to attain temperatures of about 1450 °C (30–35%), limestone decarbonation to produce clinker (50–60%), and materials handling (around 10%) [6]. Thus, it is crucial to develop new alternatives to replace PC to reduce CO_2_ emissions and conserve energy [7]. Two practices may be accepted: (i) the partial Portland cement replacement by supplementary cementitious materials to reduce energy and raw materials consumption and diminish CO_2_ emissions, and (ii) clinker production with no cement content.

It is common in the building materials sector to find traditional binders replaced by alternative secondary products such as silica fume, blast-furnace slag, fly ash, biomass, ash, red mud, and natural, and artificial puzzolans. The diversity of chemical compositions in these by-products results in new hydrated phase formation (with different chemical and physical properties found in traditional Portland cement) [8,9].

Otherwise, the alkaline activation of powders obtained from by-products based on aluminosilicates (precursors) can generate an inorganic binder when some alkaline compounds react with those precursors. The alkaline-activated binder’s interest is due to its extreme-temperature tolerance, minor permeability, stable bonding, favorable durability, attractive chemical corrosion resistance, immobilization of toxic waste, and environmental friendliness, among other benefits [10,11,12,13,14,15,16,17,18,19,20,21,22,23].

Raw materials used for geopolymer synthesis frequently come from industrial aluminosilicate wastes, natural aluminosilicate minerals, or their mixtures [9,24]. A geopolymer’s properties are obviously affected by the characteristics of raw materials (mineralogical composition, glassy/amorphous fraction, particle size distribution, particle morphology, etc.).

Large quantities of ground granulated blast-furnace slag (GGBFS) and fly ash (FA) are still available in the trading market. This fact motivates a part of the scientific community and the cement industry to focus on new geopolymer development through GGBFS and FA utilization based on their probable high hydraulic potential and monetary and atmospheric benefits [25,26,27,28].

Fly ash is a powder by-product with fine spherical particles ranging from <1 mm to more than 100 mm and produced in thermal power plants during coal burning, whose annual production is now difficult to estimate due to the general crisis of energy products (mainly oil and gas), but a few years ago, it was estimated to be about 900 million tons [29].

Alkaline activation of fly ash with high hydroxide concentrations leads to a reaction product (N-A-S-H gel) that may develop more than 60 MPa when thermally cured.

On the other hand, ground granulated blast-furnace slag is another by-product material with a high fraction of glassy phase rich in silica, alumina, and amorphous calcium, making it suitable for use as a precursor for synthesizing GP [30]. The annual production of GGBFS is about 400 million tons. GGBFS as a precursor material in geopolymer formation has been extensively investigated [31,32,33]. Since GGBFS is rich in calcium, silicon, and aluminum, the primary reaction product is a C-A-S-H-type gel, which may exist together with the geopolymer gel based on the GGBFS chemical composition, alkali activator type, and alkali concentration [31,32,33]. The C-A-S-H gel from GGBFS enhances the setting and mechanical strength characteristic of a geopolymer [34,35,36,37,38,39]. Diverse methods and mixtures of alkaline solutions with different precursors, such as sodium silicate, sodium hydroxide, sulfates, and oxides of alkali metal salts, including binary and ternary mixtures of hydroxides with sodium carbonates and sulfates, have been designed for the alkaline activation of slags [40,41].

The synergistic effect of binary ash/slag composites (with a 50/50 ratio) activated with NaOH and cured at 25 °C, which have reached compressive mechanical strength values of about 50 MPa at 28 days of reaction, has also been studied [42,43,44]. Additionally, activated ash/slag pastes with NaOH have been studied from the mineral, mechanical, and microstructural points of view. The findings have demonstrated the presence of two reaction products. A reaction product with Ca/Si = 0.8, Al/Ca = 0.6, and Si/Al = 2–3 ratio corresponds to a hydrated gel rich in Al^3+^, including Na in its structure. The other hydration product is alkaline aluminosilicate hydroxide with a three-dimensional structure [42,43,44].

Wang et al. [45] studied the synergic effect of a fly ash/slag-based geopolymer with variations of fly ash/slag ratios (20 to 60 wt.%) activated with three NaOH solutions (0.5%, 1%, and 1.5%). Experimental samples were cured at 1, 3, 7, and 28 days. It was observed that as slag content increases in a geopolymer, the compressive resistance increases, resulting in an optimal compressive strength of about 93 MPa. Deb et al. [46] studied Class F fly ash/GGBFS-based geopolymer activation at 0%, 10%, and 20% GGBFS/Class F fly ash ratio in a NaOH and Na_2_SiO_3_ solution. Compressive strength increased at higher GGBFS/Class F fly ash ratios. The highest compressive strength (51 MPa) was reached at 20% slag and 80% fly ash with a 40% NaOH and Na_2_SiO_3_ solution and cured at 20 °C. Xu et al. [47] synthesized a geopolymer based on FA and GGBFS in different grades using an activating solution formulated from a concentrated Hanford secondary waste (HSW) stimulant (5 mol/L NaOH mixed with solid binders). The optimal compressive strength reached about 52 MPa at a fly ash/GBFS mass ratio equal to 5/3.

On the other hand, many studies have demonstrated improvements of the properties of Portland cement by limestone (CaCO_3_) addition. It has been reported that limestone powders homogenize and broadly distribute clinker particle size when limestone particle sizes below 8 μm are used in a range of 10% to 40% [48].

Based on the above, there is much interest in the scientific community to study the effect of CaCO_3_ on alkaline binary compounds. Consequently, the present research evaluated the µ-limestone (CaCO_3_) addition effect on the formation of hybrid hydration gels of alkaline-base cement (slag and ash) at different curing conditions and their influence on mechanical properties.

## 2. Materials and Methods

### 2.1. Materials

The precursors used to elaborate the geopolymers were: (i) granulated blast-furnace slag from Lazaro Cardenas steel-making plant (Michoacán, Mexico), (ii) Class F fly ash (FA) from Nava thermoelectric plant (Coahuila, Mexico), and (iii) µ-limestone from mountain deposits in Nuevo León, México. Sodium hydroxide from Jalmek with 98.5% purity (Nuevo León, Mexico), bi-distilled water, and water glass (sodium silicate from Silicatos solubles, Monterrey, Mexico) were used for the alkaline solution preparation.

Before using GGBFS, FA, and limestone as precursor raw materials to prepare alkali-activated cements, a 1 h grinding process was performed on them in a vibro-energy grinding ball mill (DM 1 model, Sweco Inc.) using 80 kg of 20 mm steel balls and 5 kg of raw material charge in the mill.

Figure 1 shows the GGBFS and FA before (Figure 1a,c, respectively) and after the milling process (Figure 1b,d, respectively). After the milling process, an angular morphology in the slag was observed. Meanwhile, for the FA, the milling process directly influenced the size and surface area of the particles, thus obtaining milled fly ash (MFA) (Figure 1e).

Figure 2a–d shows the size and angular morphology of the μ-limestone particles after milling. Particles in the range of 10 to 30 μm were obtained.

Table 1 shows the physical properties of the raw materials in terms of fineness modulus and average particle size. Meanwhile, Table 2 shows the chemical composition of the raw materials. The fineness modulus test (Blaine) was carried out according to the ASTM C204-17 standard. The average particle size was evaluated using a laser granulometry technique in MICROTAC 3500 equipment (Microtract, Montgomeryville, PA, USA). The chemical analysis was performed utilizing the X-ray fluorescence (FRX) method with X EPSILON 3-X equipment (Malvern Panalytical Ltd., Malvern, Worcs, UK).

The hydraulic activity of slag was 1.23, which represents good hydraulic behavior, according to the literature [49,50,51,52].

### 2.2. Sample Preparation

FA(MFA)/slag-based binary pastes with and without µ-limestone addition were mixed, following the proportions in Table 3. The criteria used for the mass ratio selection of each mixture came from an extensive revision of the literature, trying to select unexplored gaps in parameters (mixing composition) that gave us some elements of novelty. Then, the dry pastes were activated using NaOH as the activating solution at 4 M and 8 M concentrations. The dry pastes were mixed in a Hobart mixer (A 200 model, USA) at low speed for 5 min to a well-material homogenization. The alkali solution was incorporated into the pastes at a controlled velocity for five minutes.

The L/S ratio was determined according to ASTM C230-14 to obtain good workability; this value was set between 0.22 and 0.40 (see Table 3).

After mixing, the consistency test was carried out according to the ASTM C230-14 standard. The paste was poured into the molds to create cylindrical samples of 25.4 mm × 50.8 mm, allowing it to be set at an ambient temperature for 24 h. The hardened specimens were removed from the molds and then subject to a specific curing process.

The specimens were classified and treated under three groups of curing methods: (1) saturated curing (immersed in water): water-saturated state at 100% relative humidity for 28 days; (2) controlled curing at 25 °C: permanent water spray cured at controlled temperatures of 25 °C for 28 days; and (3) controlled curing at 80 °C: water steam cured at controlled temperatures of 80 °C during the first 24 h and then set at room temperature until 28 days.

### 2.3. Test Methods

The samples were evaluated by compressive strength using an INSTRON universal mechanical testing machine (Instituto de Ingeniería Civil, Universidad Autónoma de Nuevo León, México). The evaluation was set to a 200 kg/s loading rate for 28-day-old specimens.

The final reported values were calculated as the mean value of six specimens for each paste system.

After the compression test, the samples were collected, immersed in an acetone solution for 48 h, oven-dried for 24 h to stop hydration and aging, and then crystallographiccally and microstructurally analyzed.

The phase composition of the raw material and experimental mixtures was determined using an X-ray diffractometer. For X-ray diffraction analysis, a D8 advanced Bruker model equipped with a Vantec detector (Instituto de Ingeniería Civil, Universidad Autónoma de Nuevo León, México) was used at an excitation voltage of 40 kV, current of 40 mA, scan rate of 0.05 (2θ/sec), and a 2theta (2θ) angle between 10° and 70°. The powders (mixture of raw materials without hydration process) and controlled curing mixture systems were analyzed at 28 days of age by XRD.

A scanning electron microscopy technique (SEM) was utilized to determine the microstructure of the experimental specimens using a JEOL microscope JSM-6490LV equipped with an Oxford energy dispersive detector (EDS) for qualitative and quantitative microanalysis (Instituto de Ingeniería Civil, Universidad Autónoma de Nuevo León, México).

On the other hand, the μ-limestone’s stability was studied in NaOH alkaline solutions with separate molarities of 4 M and 8 M. μ-limestone powders were added to both alkaline solutions and stirred for 24 h. Then, the resulted solution was analyzed by atomic absorption to determine if Ca^2+^ ions were dissolved. μ-limestone powders were also analyzed by X-ray diffraction after treatment in the alkaline solution.

## 3. Results

### 3.1. Mechanical Properties

Table 4 displays the compressive resistance results of all the experimental pastes studied in the present work. As observed, most of the saturated cured pastes (71%) did not reach 20 MPa, which is the lowest compressive resistance demand of an OPC 20R [53]. In this study, this criterion was used to determine the optimal mixtures in terms of compressive strength. The compressive resistance of these pastes incremented with the slag, i.e., when the amount of slag increased, the 28-days compressive resistance was increased. This phenomenon could be ascribed to the formation of C-A-S-H gels, which would reduce the porosity and densify the microstructure of the pastes’ matrix [54,55,56,57,58]. On the other hand, it can be noted that with increasing molarity of NaOH, the compressive resistance of the pastes gradually increased, which could be justified by the internal reaction of Si, Al, and Ca elements originated by the expanded breakage of the T-O-T bonds (T: Si or Al) in fly ash and Ca-O and Si-O bonds in ground granulated blast-furnace slag, incited by the high alkalinity resulting from the rising NaOH molarity [59,60]. Although it is expected that NaOH molarity would gradually increase the compressive strength, an overdose alkali solution (8 M) might increase the mixture’s water/solid ratio, contributing to a higher liquid content, which slows down the gel formation process and successive increment of poor gel reaction products [61].

The binder’s alkaline activation process would be accelerated with the L/S ratio’s diminishment due to the mixture’s consistency decrement [62]. In this case, the reaction products such as C-A-S-H- and N-A-S-H-type gels can be promptly produced with a low L/S ratio, contributing to early-age compressive resistance development [54,55].

The compressive resistance results of controlled curing at 25 °C are also shown in Table 4. As observed, all controlled curing pastes at 25 °C reached a compressive strength of >20 MPa. The compressive resistance of these pastes improved with the addition of slag. Since slag is rich in Ca, Si, and Al, the predominant reaction product was a C-A-S-H-type gel that might improve the setting and strength characteristic of alkaline-activated cement by porosity reduction and densification process of the cementitious matrix. Idawati et al. [63] studied an FA/GBFS-based geopolymer, variating fly ash/slag ratios. They claimed that the gel formation of the slag-based geopolymer was governed by a C-A-S-H-type gel. Meanwhile, fly ash-based geopolymer was controlled by a N-A-S-H-type gel.

It has been found that C-A-S-H- and N-A-S-H-type gels can exist together when slag/fly ash composites are mixed. A hybrid-type gel (N-C)-A-S-H is also identified in slag systems as a part of the calcium released by the dissolution of the slag and its incorporation into the N-A-S-H-type gel because of fly ash activation [44,64].

Regarding the mechanical strength of controlled curing pastes at 80 °C (see Table 4), most pastes did not reach 20 MPa at 28 days (62%). In this case, a high L/S ratio (high liquid content) used in a high-alkali solution (8 M) can hinder gel formation and subsequent increment of poor gel reaction products [61].

Figure 3 shows the optimal mixtures according to the compressive strength of saturated curing pastes. As shown, there was a clear improvement in compressive strength as µ-limestone was added to all the pastes evaluated in the present study. There was also a tendency where µ-limestone added to the pastes accelerated the reactions at 7 days, reaching early mechanical strength increments. The 28-days compressive resistance of all mixtures except for the high NaOH molarity (8 M) mixtures were higher than 20 MPa, accomplishing the primary demand of an OPC 20R. Nevertheless, looking at the reinforced concrete application, only mixtures with elevated slag content (80%) could achieve the standard criteria because they could reach a 28-days compressive resistance of 34 MPa. Thus, the paste with 80% slag and 20% milled fly ash with 5% of µ-limestone addition, L/S ratio = 0.25, and 4 M of NaOH could be proposed as the optimal mixture for compressive resistance.

The optimal mixtures of controlled curing at 25 °C are shown in Figure 4. In this type of curing, high increases in compressive strength were observed concerning saturated curing pastes. An improvement in mechanical strength was shown as µ-limestone was added to the 100% of GBFS. Meanwhile, no significant contribution of µ-limestone in the mechanical resistance in binary mixtures at 28 days was observed. However, the benefit was related to the early development of high strength at 7 days. On the other hand, there was a significant effect when using milled fly ash compared to that with pastes that used fly ash, since the strength values were doubled at 28 days. The surface area of the milled fly ash contributes to high reactivity, better activation, and a possible N-A-S-H-type gel formation. The paste with 60% slag and 40% milled fly ash with 5% of µ-limestone addition, L/S ratio = 0.30, and 4 M of NaOH (60S40MFA5C 4M-0.30 paste) could be suggested as the optimal mixture for compressive strength, reaching 42 MPa at 28 days.

Figure 5 shows the optimal mixtures of compressive strength of controlled curing at 80 °C pastes. There was an improvement in compressive strength as µ-limestone was added to all the pastes evaluated in these conditions. CaO content from µ-limestone might strengthen the geopolymer matrix by making an amorphous structure based on a Ca-Al-Si gel. In addition, Ca improves the compressive resistance in geopolymeric binders because when CaO content is high, porosity decreases. Additionally, the formation of amorphous-structured Ca-Al-Si gel strengthens the final reaction product [65,66,67,68].

In addition, there was a tendency where µ-limestone added to the pastes accelerated the reactions at 7 days, reaching early mechanical strength increments similar to those attained at 28 days. Taking into account the obtained results, the paste with 40% slag and 60% milled fly ash with 5% of µ-limestone addition, L/S ratio = 0.30, and 8 M of NaOH (40S60MFA5C 8M-0.30 paste) could be proposed as the most favorable mixture for compressive resistance, reaching 30 MPa at 28 days.

#### Summary Discussion of Limestone’s Effect on Mechanical Properties

As observed, the addition of µ-limestone could undoubtedly modify the mechanical responses of slag-based alkali-activated binders. As communicated in the literature, the mechanisms promoted by limestone that are favorable to the resistance development of slag-based paste are: (i) producing nucleation sites [69,70,71], (ii) increasing slag dissolution [72,73,74], (iii) improving pore refinement [69,71,72,75,76], (iv) reaching higher packing density [70], (v) enhancing reaction degree [73,75], and (vi) gel products’ intensification [73]. As was shown, the slag-based alkali-activated pastes exhibited compressive resistance higher than that of fly ash-based alkali-activated pastes provided by the C-A-S-H- or C-S-H-type gels [77,78]. The reaction intensification by µ-limestone might induce the formation of C-A-S-H- or C-S-H-type gels, probably reducing the porosity and densifying the microstructure of the pastes’ matrix [54,55,56,57,58].

The acceleration and intensification of reaction in slag-based alkali-activated pastes responds to the action of µ-limestone grains acting as nucleation sites that induce the formation of reaction products [69,70,73,74]. The nucleation sites ease the precipitation and increment of reaction products, meaning that limestone can add apparent physical modifications and minor chemical effects.

A small amount of µ-limestone might refine the pore structure [73,75,79,80]. Rashad et al. [71] concluded in their research work that a microstructural densification effect when replacing 15% slag with limestone is observed, improving compactness, causing an induced filler effect, and enhancing packing density. Using a NaOH activator and a <25% limestone addition, reduced pore volume and pore size <15 nm were found.

On the other hand, limestone in FA-based alkali-activated pastes induces a more exhaustive and speed-up reaction [81]. It has been proposed that limestone can extensively speed up gel formation since it can behave as a nucleation site for the reaction products, improving their precipitation [81,82,83].

The formation of amorphous phases (i.e., N-A-S-H) and some crystalline phases, such as quartz, mullite, hematite, and magnetite, can be found in FA-based-activated pastes activated with NaOH. When <5% of additive was added to these pastes, the same crystalline phases remained present, i.e., no new phase formation was observed except for calcite. These non-reactive phases mostly acted as fillers or micro-aggregates embedded in the matrix.

Low µ-limestone content in FA-based alkali-activated pastes can result in a more packed and denser microstructure [80,81,84,85] that eventually improves the strength of alkali-activated pastes [81]. This phenomenon might be attributed to a filler effect of unreacted particles and more gel products due to the nucleation sites generated by the limestone and other Ca-containing phases [80,81,84,85].

Meanwhile, the effect of µ-limestone on the compressive strength of binary pastes exhibits positive trends [83,86,87,88]. A C-(A)-S-H-type gel may have been present in fly ash-slag systems with a limestone addition that showed minor chemical modifications (releasing Ca^2+^) and did not present new phases [83,89]. Regarding amorphous phases, the formation of calcium-containing phases, e.g., C-S-H or calcium hydroxide, can be found along with N-A-S-H as a consequence of the limestone dissolution within FA systems [85].

In binary pastes, µ-limestone can increase strength because of the filler effect [86]. The µ-limestone addition benefited the strengthening mechanisms by pore size control and seeding-like sites simultaneously with minor chemical modifications. Additionally, the addition of µ-limestone might improve strength by a dissolution mechanism on the activator, yielding better packing [87].

The 80S20MFA5C pastes developed the highest strength among the saturated curing pastes due to possibly extensive formation of C-S-H and C-A-S-H gels induced by µ-limestone. The acceleration and intensification of the reaction in this paste could be explained by the fact that µ-limestone grains worked as seeding-like sites, facilitating the precipitation and growth of reaction products. Furthermore, a small amount of µ-limestone might have refined the pore structure. This phenomenon might have reduced the porosity and densify the microstructure of the pastes matrix, resulting in a strengthened effect.

The 60S40MFA5C pastes reached the highest compressive strength of all controlled curing at 25 °C pastes. In this case, μ-limestone might have interacted with the slag aids as a nucleation agent to C-S-H and C-A-S-H gels formation. Meanwhile, C-S-H or calcium hydroxide, N-A-S-H formation, and low N-(C)-A-S-H gels formation could be found due to the dissolution of µ-limestone within fly ash systems. Additionally, a filler effect and better packing density might have been present in this paste.

The 40S60MFA5C pastes reached the highest compressive strength of all controlled curing at 80 °C pastes. However, a hindrance in compressive strength development could be found when compared to that of the other systems, mainly by a high L/S ratio (high liquid content) used in a high-alkali solution (8 M), which negatively affected gel formation and subsequent increment of poor gel reaction products as N-A-S-H gels. However, there was a tendency where µ-limestone added to the pastes accelerated the reactions at 7 days, reaching early mechanical strength increments similar to those reported at 28 days. In this case, μ-limestone might have interacted with the fly ash as a nucleation agent for N-A-S-H gels formation.

### 3.2. SEM and X-ray Diffraction Characterization

Figure 6a,b show the 100S and 100S5C pastes’ microstructure cured at controlled conditions.

Both pastes with an L/S = 0.25 ratio reported Ca/Si values between 1.5 and 1.9. These ratios favor the C-A-S-H gels and the hydrotalcite phase formation since the Mg/Al ratio with values between 1 and 1.2. Additionally, Figure 6b shows a μ-limestone particle embedded in the cementitious matrix. The closer the EDS is performed to the particle, the more the calcium ratio increases. Furthermore, Figure 6b shows a μ-limestone particle embedded in the cementitious matrix. The closer the EDS is performed to the particle, the more the calcium ratio increases.

Figure 7a shows the microstructure of the 100MFA5C paste where the addition of μ-limestone had a significant effect on the physical properties due to the formation of hydration gels. μ-limestone could dissolve in a highly alkaline environment, which resulted in Ca(OH)_2_ particles leaching and beginning to react with the N-A-S-H gels, according to the variation of the stoichiometric ratios forming (N) C-A-S-H gels.

The amounts of Ca(OH)_2_ resulting from the dissolution of the μ-CaCO_3_ modifies the pH and can influence the decrease of MFA dissolution and N-A-S-H gel formation [90]. The dominant reaction product is the N-A-S-H gel. CaCO_3_ may show polymorphism into vaterite, calcite, and Ca(OH)_2_ by the dissolution through the gel formation process [81].

The curing temperature influenced the formation of zeolites [91], which was corroborated by the Na/Al ratios with values higher than 0.70. Zeolites do not provide higher mechanical strength but contribute to the reaction mechanisms of aluminosilicates by the temperature and molarity of the activating solution [92].

In Figure 7b, in areas where the Ca(OH)_2_ particles were closer to the interface of the N-A-S-H gels, the presence of C-S-H-type gels calculated by Ca/Si ratios >1.50 was observed.

N-A-S-H-type gels’ formation occurred at the point farther away from the μ-limestone particle interface. It should be noted that the 100MFA5C pastes showed a better reaction of the MFA particles thanks to the particle size combined with the extreme alkalinity of the activating solution and curing at a temperature of 80 °C.

Figure 8a shows the microstructural analysis by SEM of the 60S40MFA binary pastes activated with NaOH-4 M at L/S = 0.25 and immersed in water for curing. During its microstructural analysis, the formation of the characteristic hydration gels after alkaline activation for the slag composite gels (C-A-S-H) with a Ca/Si = 1.85 ratio and for the fly ash paste gels (N-A-S-H) with Si/Al > 2 ratios were found. The N-A-S-H type hydration gels’ formation occurred at the interface of the milled fly ash particles, while the hydration gels at the slag interface were C-A-S-H-type gels, shown in Figure 8b. When calcium was present in high amounts, it could be said to be a (C)-N-A-S-H gel since Ca^2+^ was not a part of the structural polymerization chain.

The joint hydration of these compounds resulted in hybrid-gel formation, such as C-S-H, C (N-A)-S-H, N(C-A)-S-H, N-A-S-H, C-A-S-H, C-A-S-H, with the type of hybrid gel mixture depending on the type of activation.

Pastes with a higher Ca/Si ratio develop a hybrid C-N-A-S-H gel in response to the Ca^2+^ freed by slag partition and its incorporation into an N-A-S-H-type gel. This gel can coexist with forming C-A-S-H gels by slag activation and N-A-S-H gels by ash activation, which is more distinguishable at earlier ages [64].

The microstructure of the ternary slag-MFA-μ-limestone pastes was analyzed in detail in Figure 9. This figure shows the stoichiometric relationships and coexistence of reported hybrid hydration gels [64,67,82]. Figure 9a shows the 60S40FA5C paste with an L/S = 0.25 ratio. Formation of hybrid gels (C-S-H and C-A-S-H) was observed through the characteristic values in the Ca/Si ratio. Si/Al ratios values >2 were detected in several areas through the microanalysis since a higher Si concentration than Al concentration was detected in the microanalysis. It could be highlighted that at the interface of both reacted particles (slag and MFA), a hybrid gel was formed with a more balanced stoichiometric ratio (Ca/Si = 1.39). This value proved the presence of a C-S-H gel. Unreacted fly ash particles were also observed. A gel with an Si/Al ratio > 3 was calculated in areas where fly ash reacted. When binary mixtures with high amounts of slag were used, the main reaction product was calcium silicate hydrates (C-S-H gel) with high amounts of tetrahedral coordinated Al [43]. In areas closest to the interface between μ-limestone and the hybrid hydration gels, a high amount of Ca^2+^ was calculated by microanalysis.

Figure 9b shows a μ-lime particle magnification where a detachment of Ca(OH)_2_ particles was observed in response to the high alkalinity of the activating solution. At the μ-limestone/paste interface, the registered Ca/Si ratio was about 1.7–2.2. Meanwhile, the N-A-S-H gels around the μ-limestone were at average values of an Si/Al ratio = 2.6.

As reported by J. Temuujin et al. [93], the addition of CaO and Ca(OH)_2_ compounds improves the mechanical characteristics of FA-based geopolymer gels. Ca(OH)_2_ is considered a more reactive additive than CaO. Ca(OH)_2_ compound addition results in the precipitation of C-S-H gels or hydrated calcium aluminosilicate phases. Ca(OH)_2_ was obtained in the present work by dissolving CaCO_3_ from the μ-limestone in an alkaline NaOH-4 M solution. This alkaline compound contributes to slag activation in binary paste mixtures.

Figure 10 shows the microstructural analysis by SEM of the 60S40MFA5C binary pastes activated with NaOH-4 M at L/S = 0.25 and immersed in water for curing. Figure 10a shows the (C)N-A-S-H gels formation by calcium addition, which was not a part of the gel structure but was present in the gel.

CaO content seems to strengthen the geopolymers, forming a structured Ca-Al-Si gel, showing a slight positive effect on the physical properties [67].

The stoichiometric ratio Si/Al = 2.8 indicated that most of the gel formed was an N-A-S-H-type gel. There was N-A-S-H-type gel formation with some areas of C-A-S-H gel formation due to the Ca/Si = 1.1 value ratio shown in Figure 10b. This figure shows a dispersed distribution of unreacted particles.

The XRD results of raw material are plotted in Figure 11 to facilitate the phase evolution analysis in alkali-activated binders. Calcite (C) (JCPDS 83-0578) and dolomite (D) (JCPDS 89-5862) phases were reported from the slag. Meanwhile, merwinite (W) (JCPDS 84-1205), mullite (M) (JCPDS 89-2644), and quartz (Q) (JCPDS 89-1962) phases were observed from the limestone. Finally, mullite (M), quartz (Q), and hematite (He) (JCPDS 33-664) phases were detected in the fly ash.

Figure 12a shows the diffractograms of 60S40MFA and 60S40FA pastes activated with NaOH-4 M at 25 °C with and without the addition of μ-limestone. These analyses helped determine whether CaCO_3_ addition influenced the reaction kinetics of hydration gel formation. Hydrotalcite formation (H) (JCPDS 41-1428) shown by the appearance of a peak between 11° and 13° (2θ) was established. C-S-H(C) gel formation was found in these pastes at 30° to 32° (2θ) peaks (JCPDS 29-0373). The angles of the “halos” match those reported in the literature [82,83].

In the diffractogram, there were also peaks corresponding to calcite (Cc) (JCPDS 83-0578) and merwinite (W) (JCPDS 84-1205) phases from the slag. For fly ash, no hydrated or unreacted phases such as mullite (M) (JCPDS 89-2644), quartz (Q) (JCPDS 89-1962), and ferrite (F) (JCPDS 80-2377) could be observed. The diffractogram of the water-saturated 60S40MFA5C between 15° and 25° (2θ) showed an amorphous halo attributed to an amorphous N-A-S-H gel reaction. The possible reaction of limestone with the alumina from the mullite may form calcium carboaluminates since this crystalline phase can occur at low angles in small proportions (10° and 15°) [83,94].

It is worth mentioning that the 60S40MFA5C paste showed 20% of the mechanical strength increment of 60S40MFA. This behavior was possibly associated with the more intensive and accelerated formation of crystalline phases of the hydrated calcium silicates attributed to the µ-limestone addition, since limestone can act as a nucleation site for the reaction products, improving their precipitation. µ-limestone content can lead to a more compact and denser microstructure that eventually improves the strength of alkali-activated pastes.

Figure 12b shows the diffractograms of the same pastes, varying the L/S = 0.25 and L/S = 0.30 ratios. No significant change in the formation of new crystalline phases was observed. However, there was C-S-H gel formation between 39° and 40° and hydrotalcite at 43° (2θ) [43]. Figure 12c shows the diffractograms corresponding to the pastes cured at controlled temperatures (pastes with 100% raw material and the addition of μ-limestone).

The analysis showed an amorphous halo between 25° and 35° (2θ), derived from the amorphous phase of the slag. The appearance of a C-S-H(C) signal at 31° (2θ) was shown in the systems with the presence of slag. In the water-saturated 100S pastes, hydrotalcite (H) formation between 11° and 13° (2θ) and C-S-H(C) gel were shown as new phases that also had high intensity between 50° and 52° (2θ). In the 100MFA5C pastes, an amorphous halo was observed between 15° and 25° (2θ). This phenomenon was attributed to the limestone addition (Cc) and the alumina present in the mullite phase from the FA. In the 100MFA5C cured at 80 °C, a zeolitic phase was observed at 37° (2θ), corresponding to a phase called chabazite (JCPDS 46-1427).

Alexander M. Kalinkin et al. 2020 [81] also reported the vitreous phase, similar to the polymorphic phase reported in this work.

Figure 12d shows the diffractogram of the powders (mixture of raw materials without hydration process) and controlled curing at 80 °C 40S60FA and 40S60MFA pastes. No different peaks in the range of 30° to 32° and 50° to 52° (2θ) were identified. The only peaks observed corresponded to the C-S-H gel from slag hydration. An amorphous halo was present between 20° and 30° due to the amorphous phase of the fly ash. The analyses showed characteristic peaks from the minerals present in the fly ash. However, this qualitative technique only showed signals of the most intense compounds.

#### Summary Discussion of Phase Evolution

XRD analysis indicated that slag as a raw material presented crystalline phases such as calcite, merwinite, and quartz [42,95]. Additionally, the glassy phase was evident in the slag [42]. It is common to find low traces of ferrite in slag; however, it was not detected in the XRD analysis, perhaps due to the low concentration as an equipment limitation (below 3%).

After slag activation, the same crystalline phases before activation were detected (calcite, merwinite, and quartz), as well as hydrotalcite and calcium silicate hydrate as new crystalline phases. The slag alkaline activation promotes the development of an Mg/Al ratio that matches the hydrotalcite phase. This formation depends on the raw material nature and the activator’s alkalinity [96]. Hydrotalcite was detected at 40° (2θ). Other investigations have also reported the hydrotalcite phase (PDF#89-0460) [75].

As was reported in the literature, the formation of the calcium silicate hydrate gel was influenced by the amount of alumina in the slag. After slag alkaline activation, calcium silicate hydrate was detected at 29.5° (2θ) [83].

The μ-limestone addition in the slag acts as a nucleating agent and a C-S-H gel precursor [75], besides a filler effect [48].

XRD analysis of fly ash as a raw material presents crystalline phases such as mullite, quartz, and hematite. The glassy phase is also evident in fly ash [97,98,99]. After fly ash activation, the same crystalline phases before activation were detected (mullite, quartz, and hematite), as well as a new zeolitic crystalline phase known as chabazite. This phase is observed at 37°. Chabazite formation depends on the curing temperature and sodium content.

On the other hand, the degree of geopolymerization in the activated fly ash is directly related to the intensity of the amorphous silica peak around 30° [98,99].

Table 5 shows the hydration gels found in the present study. These findings are based on stoichiometric ratios’ analysis by SEM. These formed gels have also been proposed and corroborated using other characterization techniques [95,100,101,102,103].

### 3.3. Chemical Reaction of CaCO_3_ in Alkaline Aqueous Solutions

Figure 13 shows the diffractogram of μ-limestone powders after mixing with a NaOH-4 M aqueous solution for 24 h by stirring. The diffractogram showed the portlandite phase at 18°, 34°, 47°, 51°, and 55° (2θ). The JCPDS cards of the portlandite (calcium hydroxide Ca(OH)_2_) and calcium carbonate (CaCO_3_) of the μ-limestone were indicated. It was found that there was a reaction between the activating solution and the addition of limestone, this addition mineral being one more ingredient for the effective activation of the binary compounds (slag and fly ash). This analysis coincides with the studies of Nicholas E. Pingitore et al. [104] on the dissolution of calcium carbonate in a sodium hydroxide (NaOH).

Nicholas E. Pingitore et al. [104] analyzed CaCO_3_ (aragonite and calcite mineralogic phases) activation in a Na(OH) solution using fine (62 to 125 μm in size) and coarse (250 to 500 μm in size) lime particles. They found CaCO_3_ in any mineralogical phase dissolved in calcium hydroxide-Ca(OH)_2_.

The product of the reaction between calcium carbonate (CaCO_3_) and sodium hydroxide (NaOH) is the formation of calcium hydroxide Ca(OH)_2_ in the form of portlandite and sodium carbonate (Na_2_CO_3_).
CaCO_3_ (s) + 2NaOH (aq) → Ca (OH)_2_ (aq) + Na_2_CO_3_ (aq)(1)

X-ray diffraction results (see Figure 13) showed a substitution of CaCO_3_ by Ca(OH)_2_ that matched Nicholas E. Pingitore and co-workers’ research.

Figure 14 shows the hexagonal morphology of portlandite-Ca(OH)_2_ caused by the CaCO_3_ reaction in aqueous and alkaline solutions. The μ-CaCO_3_ particle’s agglomeration and reacted Ca(OH)_2_ particles were also observed. The mapping showed the distribution of the compounds where a contrast between the Ca(OH)_2_, Ca (green), and Na (blue) plates was shown.

## 4. Conclusions

The µ-limestone addition on alkali-activated binders was investigated in terms of the mechanical, microstructural, and crystallographic properties. Afterwards, the most favorable mixtures were suggested. According to the results, the principal conclusions can be listed as follows:

The compressive strength of alkali-activated binders increased significantly with (i) µ-limestone addition, (ii) slag content increment, (iii) NaOH molarity, and (iv) L/S ratio reduction.

The slag-based alkali-activated pastes exhibited compressive strength higher than that of fly ash-based alkali-activated pastes. It was proposed that this mechanical characteristic was reached by reaction intensification (C-A-S-H or C-S-H gels formation), porosity reduction, matrix densification, nucleation sites availability, and pore refinement through µ-limestone addition.

Low µ-limestone content in FA-based alkali-activated pastes could result in a more packed and denser microstructure that eventually improves the strength of alkali-activated pastes. This phenomenon might be attributed to a filler effect of unreacted particles and more gel products due to the nucleation sites generated by limestone and other Ca-containing phases.

The effect of µ-limestone on the compressive strength of binary pastes exhibited positive trends. A C-(A)-S-H-type gel may be present in fly ash-slag systems. The µ-limestone addition benefited the strengthening mechanisms by pore size control and seeding-like sites simultaneously with minor chemical modifications. Additionally, the µ-limestone addition might improve strength by a dissolution mechanism on the activator, yielding better packing.

The paste with 60% slag and 40% milled fly ash with 5% of µ-limestone addition, an L/S ratio of 0.30, and 4 M of NaOH (60S40MFA5C 4M-0.30 paste) could be proposed as the most favorable mixture regarding the performance criteria of compressive strength, reaching 42 MPa at 28 days. μ-limestone might interact with the slag aids as a nucleation agent for C-S-H and C-A-S-H gels formation. Meanwhile, the formation of C-S-H or calcium hydroxide, N-A-S-H, and low polymerization N-(C)-A-S-H gels could be due to the dissolution of µ-limestone within fly ash systems. A filler effect and better packing density might also be present in this paste.

## Figures and Tables

**Figure 1 materials-16-03818-f001:**
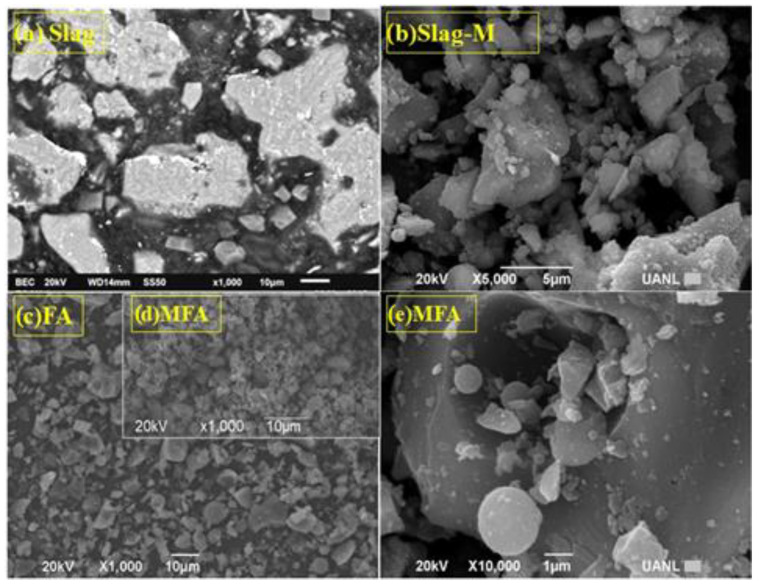
Raw materials before and after the milling process. (**a**) GGBFS before the milling process, (**b**) GGBFS after the milling process, (**c**) FA before the milling process, (**d**) FA after the milling process, and (**e**) milled fly ash (MFA).

**Figure 2 materials-16-03818-f002:**
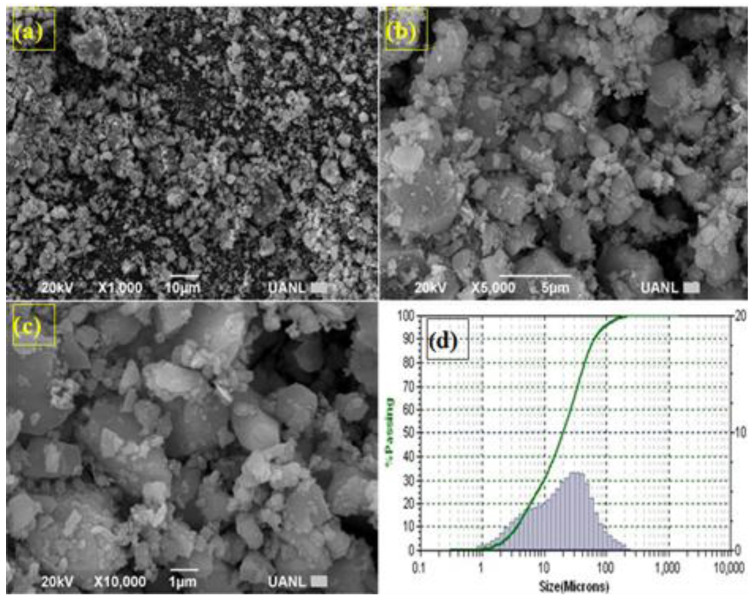
µ-limestone SEM characterization. (**a**) µ-limestone particles-X 1000, (**b**) µ-limestone morphology-X 5000, (**c**) µ-limestone angular morphology-X 10,000, and (**d**) µ-limestone particle size distribution (10–30 µm).

**Figure 3 materials-16-03818-f003:**
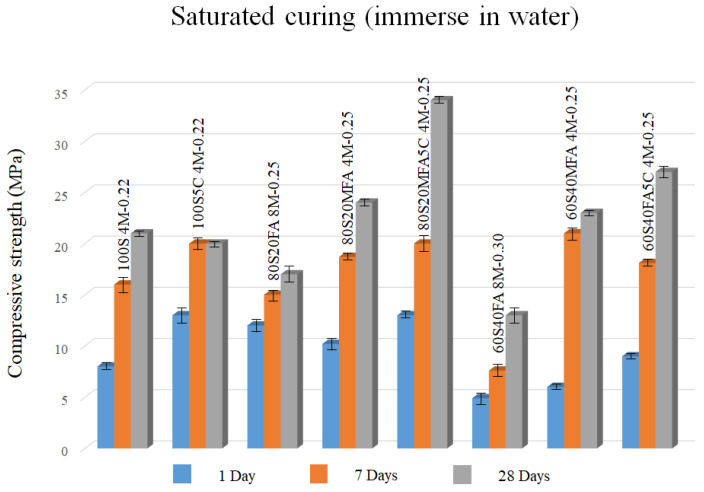
Alkali-activated pastes with 4 M and 8 M of NaOH with saturated curing.

**Figure 4 materials-16-03818-f004:**
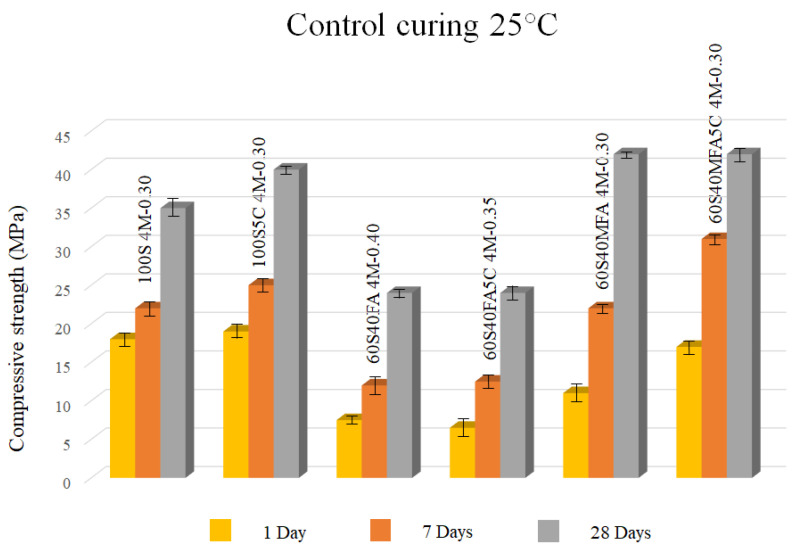
Alkali-activated pastes with 4 M and 8 M of NaOH with controlled curing 25 °C.

**Figure 5 materials-16-03818-f005:**
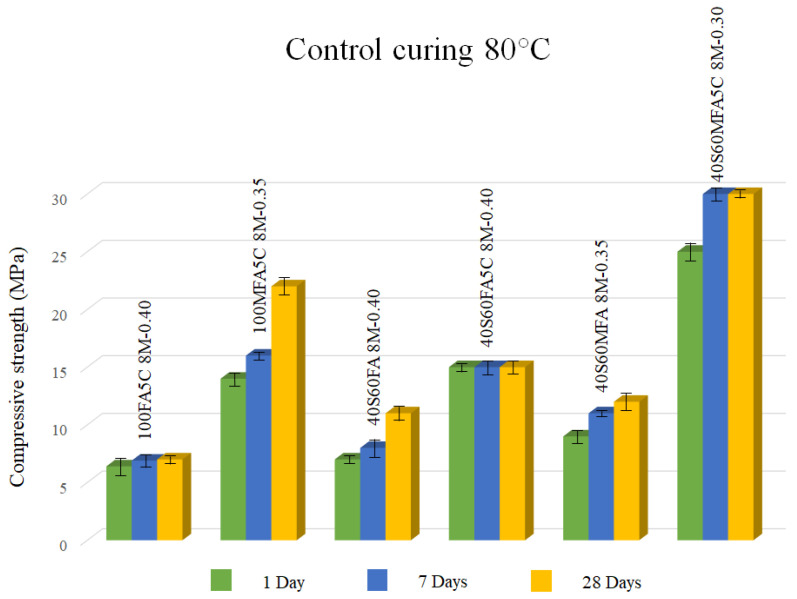
Alkali-activated pastes with 8 M of NaOH with controlled curing 80 °C.

**Figure 6 materials-16-03818-f006:**
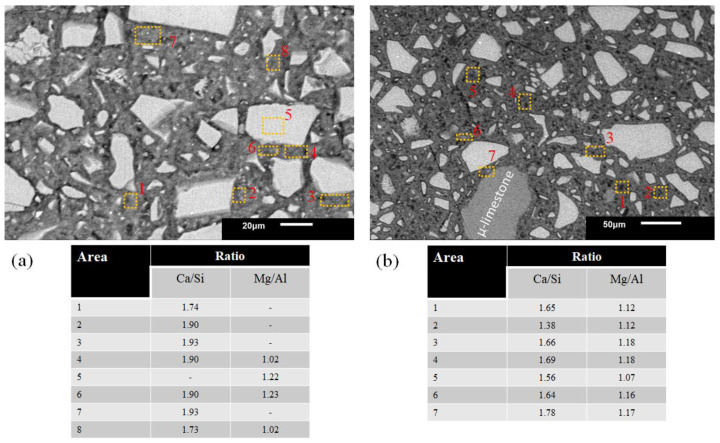
Pastes 100S and 100S5C activated with NaOH-4 M. (**a**) Hydration gels of L/S = 0.25, (**b**) distribution of µ-limestone particles in a cementitious matrix.

**Figure 7 materials-16-03818-f007:**
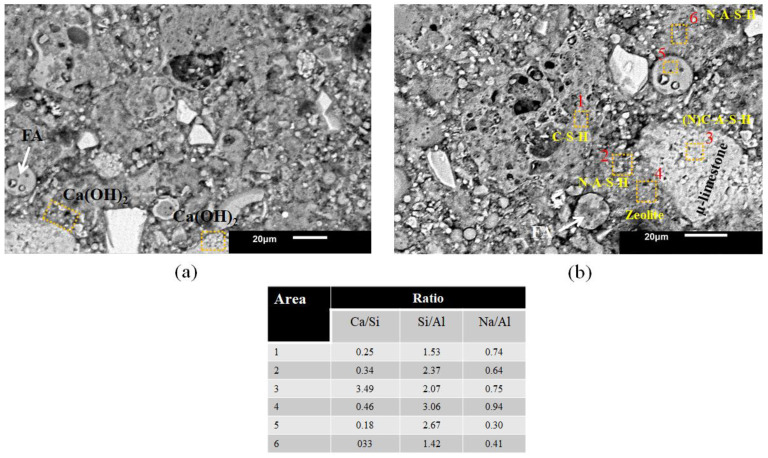
Pastes 100MFA5C activated with 8 M of NaOH. (**a**) Synthetic portlandite formation (Ca(OH)_2_), (**b**) hybrid gel formations.

**Figure 8 materials-16-03818-f008:**
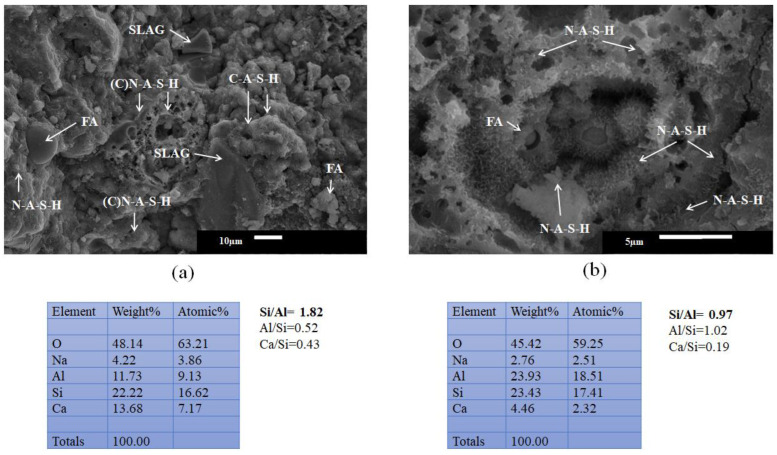
Pastes activated with 4 M of NaOH. (**a**) Hydration gels of 60S40MFA L/S = 0.25, (**b**) chemical reaction of FA with NaOH.

**Figure 9 materials-16-03818-f009:**
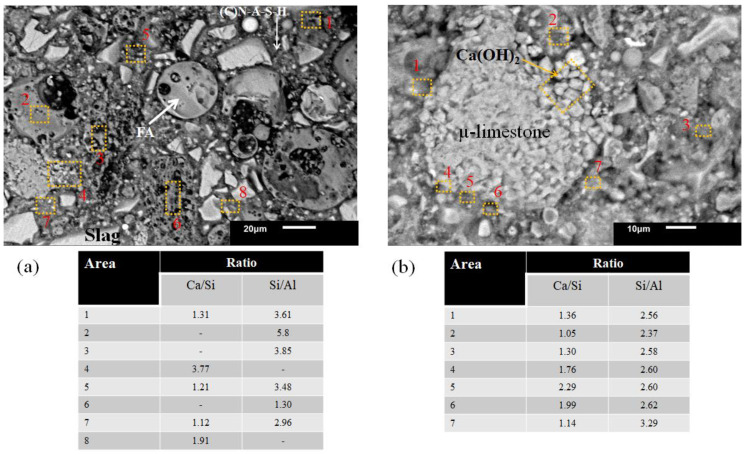
Pastes 60S40FA5C and 60S40MFA5C activated with 4 M of NaOH. (**a**) Hydration gels of L/S = 0.25, (**b**) distribution of µ-limestone particles in a cementitious matrix.

**Figure 10 materials-16-03818-f010:**
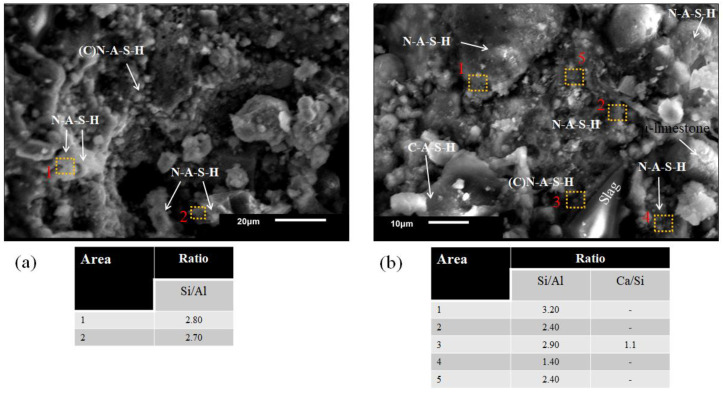
Pastes 60S40MFA5C activated with NaOH-4 M and with μ-limestone addition. (**a**) Hydration gels of L/S = 0.25, (**b**) particle distribution in a cementitious matrix.

**Figure 11 materials-16-03818-f011:**
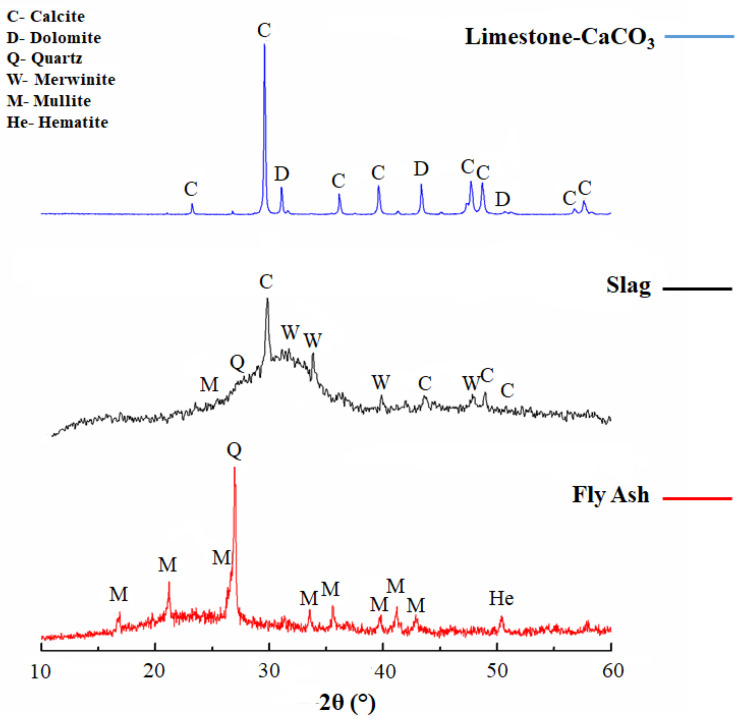
XRD of the raw materials.

**Figure 12 materials-16-03818-f012:**
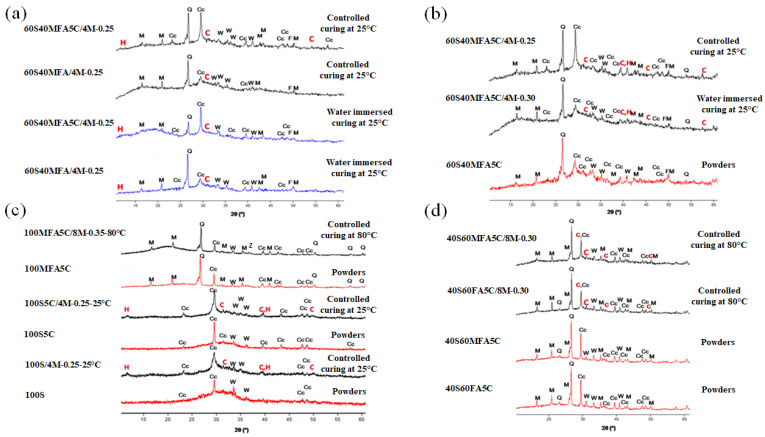
Mineralogical formation of compounds in the analyzed pastes. (**a**) Comparison between mixtures 60S40MFA with different curing methods. (**b**) Effect of µ-limestone addition in mixtures 60S40MFA5C by controlled curing. (**c**) Effect of curing temperature and µ-limestone addition in the hybrid gels’ hydration. (**d**) Effect of curing temperature in the formation of mineral compounds in binary mixtures with µ-limestone addition. (Cc = calcite, M = mullite, Q = quartz, C = calcium silicate hydrate, W = merwinite, H = hydrotalcite, F = ferrite, Z = chabazite).

**Figure 13 materials-16-03818-f013:**
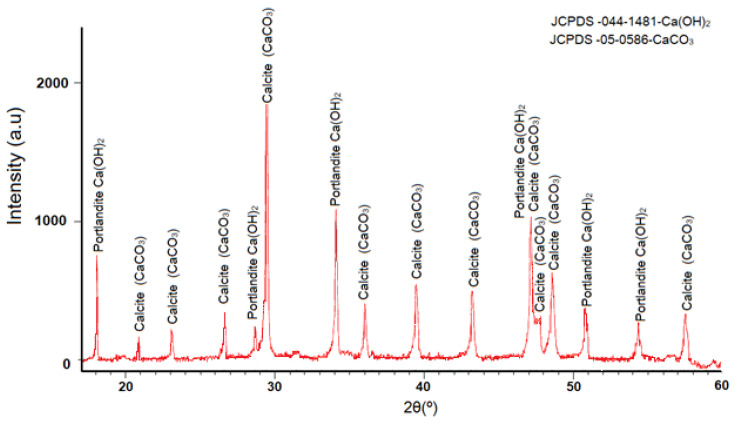
Portlandite formation by chemical reaction of NaOH in μ-limestone.

**Figure 14 materials-16-03818-f014:**
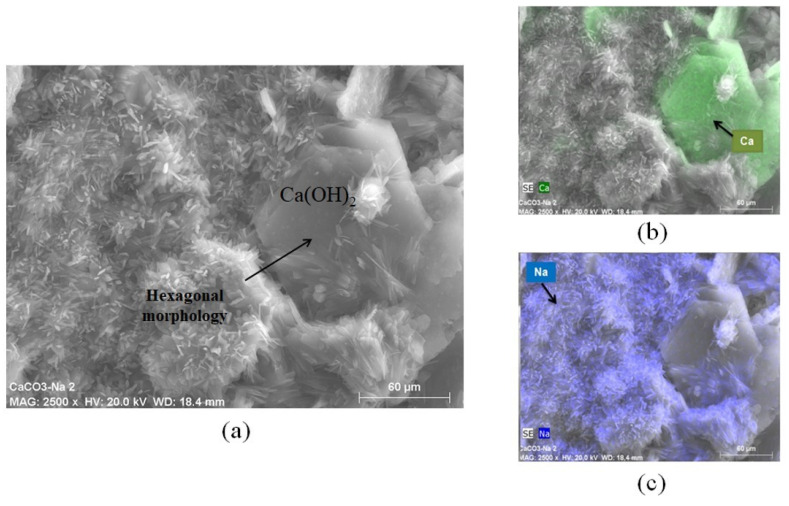
Hexagonal morphology of portlandite-Ca(OH)_2_ caused by a chemical reaction with aqueous and alkaline solutions. (**a**) Hexagonal morphology of portlandite-Ca(OH)_2_, (**b**) Ca distribution(green), and (**c**) Na plates’ distribution (blue).

**Table 1 materials-16-03818-t001:** Physical properties of the raw materials.

Raw Material	Size (μm)	Blaine (m^2^/kg)
Slag	60	4.5
Milled fly ash	30	3.8
µ-limestone	30	7.0

**Table 2 materials-16-03818-t002:** Chemical analysis of the raw materials (oxides wt.%).

Raw Material	MgO	Al_2_O_3_	SiO_2_	SO_3_	K_2_O	CaO	TiO_2_	MnO	Fe_x_O_y_	L.O.I
Slag	9.57	9.71	32.16	3.24	0.42	42.24	1.62	0.14	0.41	2.5
Fly ash	0.63	26.72	61.94	1.03	3.67	2.80	0.93	0.01	4.24	2.7
µ-limestone	1.06	0.42	1.60	0.0	0.01	97.04	0.0	0.0	0.01	-

**Table 3 materials-16-03818-t003:** Mixtures’ design.

Code	Saturated Curing (Immersed in Water)					
	Binders (wt.%)				NaOH Activator	L/S Ratio
	Slag	Fly Ash	Milled Fly Ash	µ-Limestone		
100S	100	-	-	-	4 M	0.22, 0.25, 0.30
					8 M	0.25, 0.30
100S5C	100	-	-	5	4 M	0.22, 0.25
					8 M	0.25, 0.30
80S20FA	80	20	-	-	4 M	0.25, 0.30
					8 M	0.22, 0.25, 0.30
80S20MFA	80	20	20	-	4 M	0.22, 0.25, 0.30
					8 M	0.25, 0.30
80S20MFA5C	80	-	20	5	4 M	0.25
					8 M	0.25, 0.30
60S40FA	60	40	-	-	4 M	0.30
					8 M	0.30
60S40MFA	60	40	20	-	4 M	0.25, 0.30
					8 M	0.25, 0.30
60S40FA5C	60	40	-	5	4 M	0.25, 0.30
					8 M	0.25, 0.30
	Controlled Curing 25 °C					
100S	100	-	-	-	4 M	0.25, 0.30
100S5C	100	-	-	5	4 M	0.25, 0.30
60S40FA	60	40	-	-	4 M	0.35, 0.40
60S40MFA	60	-	40	-	4 M	0.30
60S40MFA5C	60	-	40	5	4 M	0.30
	Controlled Curing 80 °C					
100FA5C	-	100	-	5	8 M	0.40
100MFA5C	-	-	100	5	8 M	0.35, 0.40
40S60FA	40	60	-	-	8 M	0.40
40S60FA5C	40	60	-	5	8 M	0.40
40S60MFA	40	-	60	-	8 M	0.35
40S60MFA5C	40	-	60	5	8 M	0.30, 0.35

**Table 4 materials-16-03818-t004:** Compressive strength results of the experimental pastes.

Code	NaOH Activator	Compressive Strength (MPa)														
		Saturated Curing (Immersed in Water)														
		L/S Ratio = 0.22					L/S Ratio = 0.25					L/S Ratio = 0.30				
		1 Day		7 Days		28 Days	1 Day		7 Days	28 Days		1 Day	7 Days		28 Days	
100S	4 M	8		16		21	6.5		11.8	18.5		3.8	6.5		14	
	8 M	-		-		-	9.6		12	18		5.5	10		13	
100S5C	4 M	13		20		20	7		12	18		-	-		-	
	8 M	-		-		-	6		17	17		11	11		11	
80S20FA	4 M	-		-		-	6.1		11	16.5		4.2	9.2		16	
	8 M	7.3		9.6		12	12		15	17		8.2	12		16	
80S20MFA	4 M	10.2		21		22	10.2		18.7	24		4.3	11		13	
	8 M	-		-		-	10		15	20		8	13		18	
80S20MFA5C	4 M	-		-		-	13		20	34		-	-		-	
	8 M	-		-		-	11		16	16		6.4	8		12	
60S40FA	4 M	-		-		-	-		-	-		4	8		11	
	8 M	-		-		-	-		-	-		4.9	7.6		13	
60S40MFA	4 M	-		-		-	6		21	23		-	-		-	
	8 M	-		-		-	8		16	18		9	13		17	
60S40FA5C	4 M	-		-		-	9		18.1	27		6	15.5		21	
	8 M	-		-		-	8.2		10	11		6.4	11		13	
		Controlled Curing 25 °C														
		L/S Ratio = 0.25				L/S Ratio = 0.30			L/S Ratio = 0.35				L/S Ratio = 0.40			
		1 day	7 days		28 days	1 day	7 days	28 days	1 day		7 days	28 days	1 day	7 days		28 days
100S	4 M	16	23		33	18	22	35	-		-	-	-	-		-
100S5C	4 M	14	24		30	19	25	40	-		-	-	-	-		-
60S40FA	4 M	-	-		-	-	-	-	-		-	-	7.5	12		24
60S40FA5C	4 M	-	-		-	-	-	-	6.5		12.5	24	-	-		-
60S40MFA	4 M	-	-		-	11	22	42	-		-	-	-	-		-
60S40MFA5C	4 M	-	-		-	17	31	42	-		-	-	-	-		-
		Controlled Curing 80 °C														
		L/S Ratio = 0.25				L/S Ratio = 0.30			L/S Ratio = 0.35				L/S Ratio = 0.40			
		1 day	7 days		28 days	1 day	7 days	28 days	1 day		7 days	28 days	1 day	7 days		28 days
100FA5C	8 M	-	-		-	-	-	-	-		-	-	6.4	6.9		7.0
100MFA5C	8 M	-	-		-	-	-	-	14		16	22	9	13		16
40S60FA	8 M	-	-		-	-	-	-	-		-	-	7	8		11
40S60FA5C	8 M	-	-		-	-	-	-	-		-	-	15	15		15
40S60MFA	8 M	-	-		-	-	-	-	9		11	12	-	-		-
40S60MFA5C	8 M	-	-		-	25	30	30	20		20	20	-	-		-

**Table 5 materials-16-03818-t005:** The hydration gels found in the present study.

Mixture Code	Suggested Gel Type [44,48,64]
100S	C-S-H C-(A)-S-H
100S5C	C-S-H C-(A)-S-H
100MFA	N-A-S-H
100MFA5C	N-A-S-H C-S-H (C)N-A-S-H
60S40MFA5C	C-A-S-H, (N,C)-A-S-H, C-(N-A)-S-H, C(N-A-S-H)
40S60MFA5C	N-A-S-H (N,C)-A-S-H C-(N-A)-S-H

## Data Availability

Data are contained within the article.

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
