# Peer review of "Microstructural and Mechanical Characteristics of Alkali-Activated Binders Composed of Milled Fly Ash and Granulated Blast Furnace Slag with µ-Limestone Addition"

_materials, 2023, doi:10.3390/ma16103818_

Round 1
Reviewer 1 Report
The papers deals with the investigation of the effect of the addition of limestone powder in mixtures composed by furnace slag/fly ash and activated with sodium hydroxide (NaOH). The study is very accurate and comprehensive from a methodological point of view. For this reason I strongly recommend his publication, as the topic is very interesting and in the paper there are several elements of novelty. However, the writing of the paper does not show the same accuracy of experimental procedures. The study contains a lot of data which need a more clear and accurate exposure, as in the actual state the reading of the paper appears very difficult and, in some parts, quite tedious. A strong and detailed English revision of the manuscript is also necessary to improve its clarity as in the present state the meaning of some sentences is not clear. So my suggestion is a careful rewriting of the manuscript and in English revision in order to improve its clarity.
Hereafter some general suggestions:
Introduction
The introduction is very long and full of details which may not be necessary. I suggest to write only necessary information for the aim of the study and probably put soem details in a comprehensive discussion in order to better focus the obtained results in the whole state of knowledge of the arguments. Furthermore, there are several sentences without a proper reference quotation. In particular: from line 113 to line 115, from line 141 to line 148, from line 204 to line 221.
Line 118: replace binder with precursor.
Materials and Methods
Line 243: I miss Figure 1d
Line 272: “water was gradually added to the pastes”. Please clarify if just water or the alkaline solution
278: the sentence “…then water-saturated cured at 100% relative humidity for 28 days” is not clear. Please write in more clear way.
Sample preparation and Table 3: please specify the criterium used for the selection of mass ratio of each mixture.
Line 281: the sentence “Water-saturated cured at 100% relative specimens” is not clear. Please re-write.
Line 306-307 The sentence “The temperature and curing type promote the leaching of Si4+ and Al3+, modifying the properties of Si4+ and Al3+ [52,102] as no sense. Please clarify.
Paragraph 3.1 need rewriting as in the present state is unclear and difficult to read. Mechanical strength plots also need a complete restyling, as in the diagrams of figure 3 and Figure 4 strengths data cannot be analysed clearly. Furthermore the list of all prepares samples with corresponding examination should be reported in Table 3 in order to clarify the meaning of sample labels in Figure 3 and 4.
Lines 512-514: “It is worth mentioning that the 60S40MFA5C paste showed 20% of a mechanical strength increment compared to 60S40MFA. This behavior is possibly associated with the formation of crystalline phases of the hydrated calcium silicates detected”. From X-ray diffraction it seems that hydrated calcium silicates are in both 60S40MFA5C and 60S40MFA samples.
Conclusion
Conclusion should be rewritten in order to better infantize the main results of the work in a more fluent discussion and not just as a list of sentences disconnected from each others. Otherwise is not clearly hFigures
Figure 10: what is the meaning of anhydrous? Please explain in the text.
SEM microphotograph in Figures 5-9 should be better designed
Tables
Table 3: I suggest a more clear illustration of mixtures.
Author Response
The response to reviewer 1 is given as an attached pdf file.

Reviewer 2 Report
As an alternative to Ordinary Portland Cement, the manuscript investigates the effect of 5 wt.% limestone powder (μ-limestone) in mixtures of blast furnace slag and fly ash activated with sodium hydroxide (NaOH) at different concentrations studied to evaluate the physical properties in the fresh and hardened state. However, the manuscript has several problems, as follows:
(1) Many studies related to limestone powder's effect on alkali-activated materials' properties (including binary alkali-activated materials). However, the manuscript must propose and solve the existing research gap. The research significance needs to be apparent, which is required to demonstrate the research novelty points of the manuscript.
(2) The primary objective of the manuscript is to investigate the effect of limestone powder on the performance of alkali-activated materials, but μ-limestone was only selected at 5 wt.%. How was the amount of its addition determined? Is there a need to investigate the effect of μ-limestone content and the addition of a control group of the conventional limestone powder? In addition, a series of studies were carried out to investigate the effects of curing conditions, slag/fly ash ratio, NaOH concentration, and fly ash fineness on the properties of alkali-activated material. How does its novelty differ from other studies?
(3) Mechanical properties (from Line 304), in which the test results are explained in a confused logical way, and their description needs to be better understood. Would this be further improved? For example, separate discussion by factor type.
(4) The manuscript needs more discussion and could have discussed in-depth how the factors and the compounding effects affect the properties of binary alkali-excited materials.
(5) What are the key conclusions of the manuscript? It is suggested not to list and highlight the major novel conclusions.
(6) Lines 135: Is there a difference between μ-lime and μ-limestone in lines 328 and unify the meaning of the symbols.
Lines 263: the mathematical expression for stoichiometric ratios is ambiguous and needs to be corrected by referring to the relevant literature.
Lines 473: why the SEM of each group in Fig. 8 is not consistent with the corresponding EDS images (magnification inconsistency).
The diagrams need to be standardized in terms of size (figure size, in-plot labels, and legends, etc.) and clarity. For example, the labels of the XRD diffraction peaks in Fig. 11 are too small.
Author Response
The response to the reviewer 2 is given as an attached pdf file.

Reviewer 3 Report
The manuscript entitled "Effect of limestone powder on binary (fly ash and blast furnace slag) alkaline activated binders" presents an interesting experimental study conducted on obtaining and characterization of alkali activated materials with µ-Limestone addition. However, the number of replicated specimens wasn’t presented, and other issues must be addressed. The paper needs major revisions before it is processed further. Some comments follow:
Title: The title is too long. Could authors replace it with a shorter formula that clearly reflects the content of the paper but also attracts the interest of readers in this field?
Introduction section
Please avoid group citations in one phrase, such as [10–24], [38–41], etc. Please discuss the highlights individually. Moreover, the introduction section is long and hard to follow. Could the author introduce a quantitative presentation of the reviewed literature? Maybe a table with the highlights (parameters and results) from the reviewed papers
Materials and Methods section
Table 2: Two types of iron oxides have been detected in these types of materials; therefore, please replace Fe2O3 with FexOy in the XRF analysis or provide the scientific proof to support your results (XRD that show the presence or absence of magnetite (Fe2O3) or hematite (Fe3O4)).
Microstructural analysis of raw materials—Please introduce figure labels to indicate those particles to which you are referring in the description.
The mixture rationale is ambiguous. What is the meaning of 100%Slag+ 5% μ-limestone. Please improve the formulation and description. Also, please add wt. next to % in the description of the mixtures (table 3).
Mix proportions – There is no rationale in chosing the amount of each constituent.
Also, considering the high content of CaO in the slag, some of the mixtures cannot be considered geopolymers but rather alkali-activated materials. See figure 1 in https://doi.org/10.1016/j.jmrt.2018.07.006 or other relevant studies.
Results section
Please move the first paragraph (lines 304–307) after the second one or cite Figure 3 in the first paragraph.
Compressive strength analysis: how many samples have been tested from each batch? Please provide the measurement values with a deviation bar.
The figures showing the microstructure of the samples could be improved by introducing some detail zones (magnification areas) and some labels indicating the areas of interest for the readers. The authors indicate some areas as containing calcium hydroxide, N-A-S-H, etc., but there is no experimental proof to support those affirmations. Please provide EDS analysis or nano-FTIR to confirm the presence of such structures.
XRD analysis: why authors considered some peaks instead of others Please evaluate all peaks that appear on the XRD spectra for each sample. The XRD spectra of the analysed sample show some clear peaks around 22,24,43, 44, 48, 55, 57 62, 2θ º etc. that weren’t considered.
Author Response
The response to the reviewer 3 is given as attached pdf file

Round 2
Reviewer 1 Report
The authors have been well addressed to my remarks, so the paper can be accepted after some further minor revisions:
1. please, check X-ray powder diffraction data and interpretations as they are not well addressed
2. I suggest to avoid the use of the term "geopolimerization" as in the case of high Ca-rich systems it is not appropriate, ut it would be better to use: "formation of a gel", "gel formation", C-A-S-H gel-like formation, and so on...
3. My previous comment: please specify the criterium used for the selection of mass ratio of each mixture. Authors answer: Thank you for your observations, the criterium used for the selection of mass ratio of each mixtures came from an exhaustive revision in the literatura, trying to select the gap in parameter unexplored (mixing composition) that gave us some elements of novelty.
Please, specify in the paper!
4. Figure 10: what is the meaning of anhydrous? Please explain in the text.
Of course "anhydrous" is without water, but I suggest to avoid the use of anhydrous and hydrated mixture system to indicate samples cured in different conditions (i.e. water-saturated conditions and temperature-controlled ones)
Author Response
The responses to the reviewer 1 is given as an attached pdf file.

Reviewer 2 Report
The manuscript has been considerably revised, and its aspects have been greatly improved. However, there are still some minor issues in the manuscript that need to be revised, as follows:
(1)Lines29: ")"
(2)Table 4 could perhaps be considered for removal, as the contents of the table are shown in Figures 3-5.
(3)Ground fly ash abbreviated as MFA is not reasonable
(4)Lines158: Figure name and figure are not on the same screen
(5)Lines 523: It is suggested that the titles of the sub-figures be labeled separately in Figure 13 to facilitate the reader's understanding. In addition, the atomic absorption analysis of the calcium ion part of the content is not very clear was found.
In conclusion, it is suggested that the manuscript can be directly accepted after minor revision again.
Author Response
The responses to the reviewer 2 is given as an attached pdf file.

Reviewer 3 Report
Dear authors,
You have done a good job in revising the paper, considering the previous recommendation. Further, there are some improvements that must be considered in order to make the paper clear for the readers and to assure the accuracy of the results.
1. Please provide the XRD results for the raw materials and clearly indicate the phase evolution or transformation after activation.
2. The authors stated that "Portlandite formation is caused by the chemical reaction of NaOH in -limestone,", but this cannot be confirmed considering the provided results. Moreover, where is Na in Portlandite? Please check your results and introduce relevant citations to support your affirmations.
3. The entire XRD section is based on the author's assumptions; please improve the discussions by considering relevant literature.
4. The conclusion section is way too long. Please improve it.
5. The arrows from the microstructural images are not accurate. Please check the indicated zones. The authors indicate some zones as FA, but there are no FA particles in the area, etc.
6. What is the meaning of K in the tables from Figure 8?
Best regards,
Author Response
The responses to the reviewer 3 is given as an attached pdf file.

Round 3
Reviewer 3 Report
The authors considered all my recommendations and revised the manuscript accordingly. The paper can be processed further toward publication.
Best regards,